Additions to the phylogeny of colubrine snakes in Southwestern Asia, with description of a new genus and species (Serpentes: Colubridae: Colubrinae)

Rajabizadeh Mahdi khosro.rajabizadeh@gmail.com 1 2
Pyron R. Alexander 3
Nazarov Roman r_nazarov@mail.ru 4
Poyarkov Nikolay A. 5
Adriaens Dominique 6
Herrel Anthony 2 6 7
1 Department of Biodiversity, Institute of Science and High Technology and Environmental Sciences, Graduate University of Advanced Technology , Kerman , Iran
2 Département ‘Adaptations du vivant’, Museum national d’Histoire naturelle , Paris , France
3 Department of Biological Sciences, the George Washington University , Washington, D.C. , United States of America
4 Zoological Museum, Moscow State University , Moscow , Russia
5 Department of Vertebrate Zoology, Faculty of Biology, Moscow State University , Moscow , Russia
6 Department of Biology, Evolutionary Morphology of Vertebrates, Ghent University , Ghent , Belgium
7 Department of Biology, Functional Morphology, University of Antwerp , Antwerp , Belgium
Pie Marcio
Electronic publication date: 2020 Apr 21
Publication date: 2020
Volume: 8
Electronic Location ID: e9016
Received 2019 Oct 22; Accepted 2020 Mar 28
Copyright: ©2020 Rajabizadeh et al.
Copyright year: 2020
Copyright holder: Rajabizadeh et al.
License: This is an open access article distributed under the terms of the Creative Commons Attribution License, which permits unrestricted use, distribution, reproduction and adaptation in any medium and for any purpose provided that it is properly attributed. For attribution, the original author(s), title, publication source (PeerJ) and either DOI or URL of the article must be cited.
License URL: https://creativecommons.org/licenses/by/4.0/

Keywords: Persiophis, Phylogeny, Supermatrix, Squamata, Iran, Hierophis, Muhtarophis, Dolichophis, Coluber

Funding: French Embassy in Tehran Russian Science Foundation 19-14-00050 US NSF DEB-1441719 Specimen storage and examination was completed with a postdoc fellowship from the French Embassy in Tehran to Mahdi Rajabizadeh, financial support from the Russian Science Foundation (RSF grant No. 19-14-00050, sample collection, data analysis) to Nikolay A. Poyarkov, and by US NSF grant DEB-1441719 to R. Alexander Pyron. The funders had no role in study design, data collection and analysis, decision to publish, or preparation of the manuscript.

==============================
Reptiles are still being described worldwide at a pace of hundreds of species a year. While many discoveries are from remote tropical areas, biodiverse arid regions still harbor many novel taxa. Here, we present an updated phylogeny of colubrid snakes from the Western Palearctic by analyzing a supermatrix of all available global snake species with molecular data and report on the discovery of a new genus and species of colubrine snake from southeastern Iran. The new taxon, named Persiophis fahimii Gen. et sp. nov., is nested within a clade containing Middle Eastern and South Asian ground racers (Lytorhynchus, Rhynchocalamus, Wallaceophis, and Wallophis). This species has a derived morphology including an edentulous pterygoid and occurrence of short and blunt teeth on the palatine, maxillae and dentary bones, an elongated snout and a relatively trihedral first supralabial scale that is slightly bigger than the second, and elongated toward the tip of rostral. We also report on the osteology and phylogenetic placement of several poorly studied colubrines: Hierophis andreanus (reassigned to Dolichophis) and Muhtarophis barani.

Introduction

The family Colubridae is the most speciose group of snakes, inhabiting a diverse array of ecosystems worldwide except Antarctica and some remote oceanic islands (Vitt & Caldwell, 2013). While at least 1,959 species of colubrid snakes have been described (Uetz, Freed & Hosek, 2020), there are undoubtedly many new species remaining to be discovered. Colubrid phylogeny has been recently studied at higher levels (Lawson et al., 2005; Nagy et al., 2004; Pyron et al., 2011; Vidal et al., 2007; Zaher et al., 2019) and with species-level sampling (Figueroa et al., 2016; Pyron, Burbrink & Wiens, 2013; Zheng & Wiens, 2016), but many nodes remain poorly supported. One of the possible reasons is the absence of unsampled or undescribed taxa, hampering the estimation of a complete phylogeny (see Pyron, Burbrink & Wiens, 2013). This emphasizes the importance of studying the biodiversity of colubrid snakes, not only from a conservation point of view (Böhm et al., 2013), but also to shed light on the phylogeny of the whole group.

In phylogenetics, extensive sampling may increase accuracy (Zwickl & Hillis, 2002). New methods and data may help in studying the biodiversity of rare colubrids, whether it be the discovery of new species, or the placement of enigmatic taxa. Thus, densely-sampled supermatrices of all available gene sequences are desirable to improve phylogenetic estimations (Pyron et al., 2011), both to cement the placement of sampled species as well as the recognition of new taxa. Furthermore, integrative taxonomy can help in accommodating different data sources and provide new avenues for delimiting species using different organismal properties (Padial et al., 2010).

Of Western Palearctic colubrid snakes in southwestern Asia (including Iran and adjacent countries), the phylogenetic relationships of a considerable number of rare or narrowly endemic taxa are unknown (Rajabizadeh, 2018). Here, we present an updated phylogeny of colubrid snakes from the Western Palearctic by analyzing a supermatrix of all available global snake species (Figueroa et al., 2016). To this matrix, we have added data from two rare and poorly known species from Iran. The first is Andreas’ Racer, Hierophis andreanus (Werner, 1917), an endemic colubrid snake from Iran with ambiguous phylogenetic placement (Figueroa et al., 2016; Schätti & Monsch, 2004), currently classified in the genus Hierophis (Wallach, Williams & Boundy, 2014). The second is a previously unknown taxon, discovered by us during a 2008 field survey in southeastern Iran, which shows unique morphological adaptations and represents a new genus and species of colubrid snakes. In addition to the molecular phylogeny, we provide osteological data for consideration in taxonomic evaluations.

Materials and Methods

Nomenclatural acts

The electronic version of this article in Portable Document Format (PDF) will represent a published work according to the International Commission on Zoological Nomenclature (ICZN), and hence the new names contained in the electronic version are effectively published under that Code from the electronic edition alone (see Articles 8.5–8.6 of the Code). This published work and the nomenclatural acts it contains have been registered in ZooBank, the online registration system for the ICZN. The ZooBank LSIDs (Life Science Identifiers) can be resolved and the associated information can be viewed through any standard web browser by appending the LSID to the prefix http://zoobank.org/. The LSID for this publication is as follows: urn:lsid:zoobank.org:pub:4EAACC14-8FC8-46B9-830C-4AEC8A42A562. The online version of this work is archived and available from the following digital repositories: PeerJ, PubMed Central and CLOCKSS.

Specimen Collection

During one month of fieldwork in southern and western Iran (Fig. 1) in May 2008, two specimens of rare colubrid snakes were collected by R. Nazarov and M. Rajabizadeh: a sample of Hierophis andreanus, and a specimen of Colubridae Gen. et sp. nov. superficially resembling snakes of the genera Rhynchocalamus or Lytorhynchus. Voucher specimens were preserved and deposited in the International Center for Science, High Technology and Environmental Sciences Herpetological Collection (ICSTZM), Kerman, Iran, along with tissue samples (muscle tissue, preserved in 100% ethanol) for molecular phylogenetic analysis. The skulls of both specimens were examined using micro-CT scanning. Specimen information are as follows:

Figure 1 Distribution map of Persiophis fahimii. Gen. et sp. nov. (star) and Hierophis andreanus (circle) in Iran.

Localities of Hierophis andreanus are based on Rajabizadeh (2018). Arrows indicate the locality of Hierophis andreanus specimens for which genetic data were included in the molecular phylogenetic analysis, along with Persiophis fahimii Gen. et sp. nov.

1. Hierophis andreanus, collected from around Darreh Shahr City, western Iran, Ilam Province (Fig. 1), preserved in 75% ethanol and cataloged as ICSTZM.7H.1154.

2. Colubridae Gen. et sp. nov., ICSTZM.7H.1151, collected from around Orzueeyeh City, southern Iran, Kerman Province (Fig. 1), preserved in 75% ethanol.

Field work, including collection of the samples and animals in the field, was performed outside of any protected area, in the framework of a project contract signed by International Center for Science, High Technology and Environmental Sciences, Kerman, Iran (contract number 1.87, issued at 11.04.2008). The contract grants permission to collect the reptile samples outside of any protected area administered by the Department of the Environment (specified in http://www.doe.ir/), which need extra permissions. Specimen collection protocols and animal operations followed the Institutional Ethical Committee of International Center for Science, High Technology and Environmental Sciences, Kerman, Iran (certificate number 1.87-1).

Molecular Phylogeny

We used the species-level supermatrix from Figueroa et al. (2016), which is currently the largest such dataset available for snakes (a dataset comprising 1,745 taxa representing 1,652 snake species and seven outgroup taxa, and 9,523 base pairs from 10 loci, accessible in https://doi.org/10.1371/journal.pone.0161070.s002). First, we removed the chimeric representative of “Lytorhynchus diadema” and added several newly-sequenced specimens from this clade derived from recent publications (Table 1). For our two new samples, we then added sequences for the mitochondrial genes 12S, ND4, and CYTB (sequencing details as in Pyron et al., 2011), accessioned in GenBank (Table S1). For the new colubrid taxon, we also sequenced the nuclear genes BDNF, CMOS, NT3, and RAG2 (accessions in Table S1).

Table 1 Additional specimens added to the matrix of Figueroa et al. (2016).

Species	Locality	Voucher	Reference	
Muhtarophis barani	–	ZMHRU2014/60-5	Tamar et al. (2016)	
Wallaceophis gujaratensis	–	NCBS HA-105	Mirza et al. (2016)	
Wallophis brachyura	–	–	Mirza & Patel (2018)	
Lytorhynchus maynardi	Iran	MVZ234499	Tamar et al. (2016)	
Lytorhynchus maynardi	Pakistan	MVZ248463	Tamar et al. (2016)	
Lytorhynchus gaddi	Iran	MVZ234500	Tamar et al. (2016)	
Lytorhynchus diadema	Oman	CN4093	Tamar et al. (2016)	
Lytorhynchus diadema	Morocco	IBES1329	Tamar et al. (2016)	
Lytorhynchus diadema	Egypt	SPM002589	Tamar et al. (2016)	
Rhynchocalamus satunini	Iran	CAS228723	Tamar et al. (2016)	
Rhynchocalamus satunini	Turkey	ZMHRU2015/0	Tamar et al. (2016)	
Rhynchocalamus arabicus	Oman	CN4780	Tamar et al. (2016)	
Rhynchocalamus dayanae	Israel	TAU.R17093	Tamar et al. (2016)	
Rhynchocalamus melanocephalus	Israel	HUJ.R22054	Tamar et al. (2016)	

We used the same partitioning and analytical strategy as Figueroa et al. (2016). With their model-partitions file (by gene and codon), we estimated the Maximum-Likelihood phylogeny using IQ-TREE (Nguyen et al., 2014), under a partitioned model (Chernomor, Von Haeseler & Minh, 2016) with the best partitioning scheme chosen by AIC (Kalyaanamoorthy et al., 2017) and support estimated using 1,000 ultra-fast bootstraps (Hoang et al., 2017) and the branch-specific SHL-aLRT statistic (Anisimova et al., 2011). Following recommendations from the developers of IQ-TREE and previous studies (Anisimova et al., 2011; Pyron et al., 2011), we interpret clades with UF>95 and SHL>80 to indicate strong support. While we re-estimated the entire snake phylogeny, we only report results from the clade of interest containing our focal taxa and other Western Palearctic Colubrids.

Osteology

The skull osteology of Hierophis andreanus was compared with that of closely related genera, including Hierophis (H. gemonensis—MNHN 1937-454) and (H. viridiflavus—MNHN 1967-79, 1869-806), (loaned from the Muséum national d’Histoire naturelle in Paris), Dolichophis and Eirenis from published reports (Hosseinian Yousefkhani & Rajabizadeh, 2014; Mahlow et al., 2013). Additionally, we gathered novel osteological observations regarding Colubridae Gen. et sp. nov., for a thorough description of the new taxon. The micro-CT scans of the heads of two snake specimens were performed at the Centre for X-ray Tomography of Ghent University (Vlassenbroeck et al., 2007). The setup was a transmission head of a dual-head X-ray tube (Feinfocus FXE160.51) and an a-Siflat panel detector (PerkinElmer XRD 1620 CN3 CS). The focal spot size was 900 nm at a tube voltage of 130 kV for high resolution. Number of projections and voxel size of the scanned specimen is presented in Table S2. Exposure time was 2 seconds per projection, resulting in a 360 degree output CT Scan. The raw data were processed and reconstructed using the in-house CT software Octopus (http://www.octopusreconstruction.com) (Vlassenbroeck et al., 2007) and rendered using Amira V. 5.4.1 (Mercury Systems of Visage Imaging GmbH). The CT-rendered images were color coded to distinguish separate ossified units, where stiff and rigidly interconnected bones were given a single color.

External morphology and comparisons

Morphological examination was performed following to Dowling (1951). Snout-vent length (SVL) and tail length (TailL) were measured using a body ruler; other measurements to the nearest 0.1 mm using a Mitutoyo digital caliper. Pileus length was measured as the maximum distance from tip of snout to posterior margin of parietals. Dorsal scales rows were counted at the level of one head length posterior to the head (anterior body), midbody and one head length anterior to the anus (posterior body). Comparative morphological data were extracted from the literature (Boulenger, 1898; Bourgeois, 1968; Broadley, 1994; Broeckhoven & Du Plessis, 2017; Chan-Ard, Nabhitabhata & Parr, 2015; Chippaux & Jackson, 2019; Das et al., 2019; Gans, 1954; Kardong, 1979; Kharin & Akulenko, 2008; Mahlow et al., 2013; Marx, 1959; Mirza & Patel, 2018; Mirza et al., 2016; Nguyen et al., 2020; Poyarkov, Nguyen & Vogel, 2019; Rajabizadeh, 2018; Saleh & Sarhan, 2016; Schätti, 1985; Schätti, 1987; Tsai & Mao, 2017; Utiger, Schätti & Helfenberger, 2005; Wade, 2008; Wagner & Böhme, 2007; Wallach, Lanza & Nistri, 2010; Wang, Shi & Guo, 2019).

Results

Phylogeny

The ML tree (see Fig. 2) is overall highly similar to many recent estimates of colubroid snake phylogeny (Figueroa et al., 2016; Pyron et al., 2011; Zaher et al., 2019), with a few major exceptions highlighted in part by our new sampling. Our results confirm the occurrence of a monophyletic group (UF/SHL = 93/97) of Western Palearctic colubrids including 17 genera and Colubridae Gen. et sp. nov. However, phylogenetic placement of all genera within this clade is not fully resolved, due to low support values for some nodes. Our results strongly support a sister-group relationship of Muhtarophis barani and Scaphiophis albopunctatus (83/97), which together form a clade that is the sister group to all above mentioned genera of Western Palearctic and South Asian colubrids with high support (-/95). There are two main subclades of western Palearctic and South Asian colubrines that we highlight for further attention.

Figure 2 Phylogenetic relationships of Western Palearctic Colubrids.

(A) Resulting topology and UF/SHL-support values from reanalysis of the matrix of Figueroa et al. (2016) with additional colubrines from SW Asia; (B) phylogenetic relationships within the clade of Western Palearctic colubrids. The species Hierophis andreanus is nested within Dolichophis with strong support, while Persiophis fahimii Gen. et sp. nov. forms a distinct lineage as the sister group to Rhynchocalamus and clearly represents a new genus and species.

The first involves a small radiation of colubrine ground-snakes endemic to southwestern Asia (including Colubridae Gen. et sp. nov.), and the second involves colubrines from the Old-World racer lineage. The first subclade is strongly supported (100/100), and includes Colubridae Gen. et sp. nov. as the sister lineage to a weakly supported clade containing Rhynchocalamus, Wallophis + Wallaceophis, and Lytorhynchus. The monophyly or sister relationship of each of those genera is strongly supported by at least one measure. However. the phylogenetic relationships between the latter genera are only moderately supported (85/70 and 75/68). The genetic distance of the new colubrid lineage from the remaining taxa combined with its morphological distinctiveness necessitates a novel generic assignment.

The second sub-clade is divided into several groups. In the first lineage, Mopanveldophis zebrinus is the sister group (89/100) to the genera Bamanophis and Macroprotodon. This lineage is sister to other remaining genera in the Western Palearctic colubrine clade (100/100). The second lineage contains Hemorrhois as the sister group to the genera Spalerosophis and Platyceps with strong support (98/100). Relationships in the latter genus are poorly resolved (Fig. 2). We also note that recent cranial osteological and molecular phylogenetic data from Argyrogena fasciolata place it as the sister lineage of Platyceps (Das et al., 2019) in this clade.

Among the remaining lineages, the Western Palearctic whip snakes (genera Hierophis and Dolichophis), the Slender Racer (Orientocoluber), Hierophis andreanus, and dwarf snakes of the genus Eirenis are confirmed in our tree as a monophyletic group (100/100). The genus Hemerophis is weakly supported as the sister lineage to this entire clade. The clade comprises one lineage including the genera Orientocoluber and Hierophis (sister relationship strongly supported; 98/100), and a second lineage including the genera Dolichophis and Eirenis. Although the monophyly of each of the latter genera is confirmed, their sister group relationship is not supported by UF, only SHL (99). Furthermore, Hierophis andreanus is placed within the genus Dolichophis as the sister group to D. jugularis with strong support (91/98), and we thus formally reassign H. andreanus to Dolichophis. Based on our tree, monophyly of the genus Dolichophis is confirmed (100/100), with a basal divergence within the genus between a sub-group comprising D. schmidti and D. caspius (100/100), and another sub-group of D. jugularis and Dolichophis andreanus comb. nov.

Systematics

Phylogenetic results indicate that Colubridae Gen. et sp. nov. is nested within the subfamily Colubrinae, and we estimate moderate support for a sister group relationship with clade including snakes of the genera Rhynchocalamus, Wallophis, Wallaceophis, and Lytorhynchus. The Colubridae Gen. et sp. nov. is thus distantly diverged from all other colubrid snakes for which sequence data is available. From an osteological point of view, Colubridae Gen. et sp. nov. lacks teeth on the premaxilla and proteroglyphous or solenoglyphous teeth on the maxilla, a coronoid bone in the mandible, girdle or limb elements, and valvular dorsal nostrils (see Osteology, below), what confirms its assignment to the family Colubridae (Vitt & Caldwell, 2013). The occurrence of a broad articulation between snout bones and the lack of numerous and closely-set teeth, as well as the fact that the specimen was found on an arid mountain side and does not have an aquatic or semiaquatic lifestyle confirms that Colubridae Gen. et sp. nov. belongs to the subfamily Colubrinae rather than the related subfamily Natricinae (Vitt & Caldwell, 2013; Zaher et al., 2012). Although the occurrence of an edentulous pterygoid is observed in the genera Dasypeltis, Lytorhynchus and Rhynchocalamus as well (Avcı et al., 2015; Gans, 1952; Gans, 1954; Leviton & Anderson, 1970), the combination of osteological traits of Colubridae Gen. et sp. nov. is unique within the subfamily, indicating that this single specimen represents a new genus as well as a new species of colubrid snakes, which are described herein as follows:

Persiophis fahimiiGen. et sp. nov.	
(Figs. 3–5 and Fig. S1)	

Holotype. Adult female, ICSTZM.7H.1151 (field number: RAN 2948). Iran, Kerman province, 19 km NW of Orzueeyeh City, 1350 m ASL; coll. R. Nazarov, May 2008 (Figs. 3–5 and Fig. S1). The specimen originally preserved in 96% ethanol and then moved to 75% ethanol for long term preservation. Since the authors are concerned about the conservation of the species, the exact geographic coordinates of the type locality are not given herein, but can be obtained upon request from the authors.

Figure 3 The holotype of Persiophis fahimii. Gen. et sp. nov., live specimen in situ.

Details of head scalation in close-up (A) lateral, (B) ventral and (D) dorsal views; (C) lateral view of the fore body, and (E) dorsal view of the whole body. Photos by Roman A. Nazarov.

Figure 4 Micro-CT reconstruction of the skull of Persiophis fahimii Gen. et sp. nov.

(A) Lateral, (B) dorsal, (C) ventral and (D) sagittal views of the skull of the holotype. Visualization by Mahdi Rajabizadeh.

Figure 5 Habitat of Persiophis fahimii. Gen. et sp. nov. at the type locality in vicinity of Orzueeyeh City, Kerman Province, Southern Iran.

(A) Macrohabitat, arrow indicates the place where the snake was collected; (B) microhabitat at the site of collection of the type specimen. Photos by Mahdi Rajabizadeh.

Etymology. The genus name is a latinized noun in masculine gender derived from the Greek words “Persi-” (Persís) = Persia (old name of Iran) and “ophis” = serpent. The species is named after Dr. Hadi Fahimi, a young naturalist and herpetologist who dedicated his life to studying the biodiversity and conservation of reptiles and mammals of Iran. As a young nature lover, Hadi joined the rangers of the Department of Environment in Kerman province for two years and served partly in Khabr National Park where is close to the type locality of Persiophis fahimii. He was a PhD student in IAU, Tehran, studying on the conservation of black bears in southeastern Iran, but sadly passed away in an aircraft crash in Dena Mountain in central Zagros in February 2018. We suggest the common name “Fahimi’s Ground Snake” in English for the new species.

Diagnosis. For the genus and species, Persiophis fahimii is distinguished within the subfamily Colubrinae by a combination of distinct osteological characters, including the occurrence of vestigial teeth on the palatine; a thin, edentulous pterygoid; short and blunt teeth on the maxillae and dentary, occurrence of edentulous parts on the anterior and middle region of the maxillae; a fully fused basioccipital and basisphenoid; the occurrence of a highly oblique quadrate bone attached to the posterior tip of a somewhat elongated supratemporal. The genus and species are also distinguished within the subfamily Colubrinae by a combination of morphological characters, including an elongated snout; occurrence of a rostral scale that is visible from above and wedged between the internasals; a relatively trihedral first supralabial that is slightly bigger than the second and elongated toward the tip of rostral; 15 longitudinal rows of dorsal scales on midbody; and an edentulous pterygoid.

Comparisons. The above mentioned anatomical traits are in contrast to those observed in the genus Rhynchocalamus, including a small, thin, down and backward directed premaxilla; a broad, edentulous pterygoid; relatively elongated, posteriorly curved teeth on the maxillae and dentary; a closed suture between basioccipital and basisphenoid; a short and nearly vertical quadrate bone on each side of cranium, and a broad attachment surface for a short supratemporal (Avcı et al., 2015). Also, Persiophis fahimii differs from Rhynchocalamus and Lytorhynchus in having maxillae that anteriorly and medially are edentulous and in between, bear small and vestigial teeth except for the last two, in contrast to relatively elongated, posteriorly curved teeth over most of the maxillar length in Rhynchocalamus and Lytorhynchus (Avcı et al., 2015; Leviton & Anderson, 1970). Persiophis differs from Dasypeltis in having smooth edges on the anterior frontal and posterior nasals (compared to small premaxilla and a serrated anterior free edge of frontals and posterior edges of nasals) (Gans, 1952).

Description of the holotype. Body and tail slender and elongate. Head small, oblong-shaped, slightly distinct from neck; snout elongated. Snout-vent length 380 mm, tail length 115 mm, head length 12.9 mm, head width 7.6 mm, pileus length 9.6 mm, parietal length 4.5/4.8 mm, (right/left), frontal length 3.0 mm, frontal width 2.2 mm, prefrontal suture length 1.2 mm, eye diameter 1.3 mm, distance between nostrils, 2.2 mm, interocular distance 3.2 mm.

Head scalation. Tip of rostral scale visible from above and wedged between the internasals. Internasal slightly shorter in length than the prefrontal scale. Width of the frontal scale is smaller than its length, shorter than parietals. Supraoculars are smaller in length and width than the frontals. Parietals elongated, medial suture between scales crooked-shaped giving an asymmetrical appearance. Nasal scale elongated and rectangular, the nostril situated upward, approximately mid length of the nasal. Loreal is small, longer than wide. 8/8 (hereafter values given in right/left order) supralabials , the first supralabial is relatively trihedral, slightly bigger that the second scale and elongated toward the tip of rostral, the fourth and fifth bordering the eye. A single presubocular on each side of the head, 1/2 postoculars; 3/2 anterior and 3/3 posterior temporals. 8/8 infralabials bordering the mouth on each side of the head, the first through fifth bordering the anterior genials. On the underside of the head, the mental small and triangular. Anterior genials small, in contact with each other, obliquely elongated towards the border of mouth, median suture between the anterior genials about the length of mental scale. Posterior genials contacting each other, elongated and larger than the anterior genials, median suture between them slightly more than twice the length of suture between the anterior genials.

Body scalation. Dorsal scales smooth, having a single apical pit. Dorsal scales at the anterior body, midbody and posterior body are in 19, 15, and 15 longitudinal rows, respectively. Dorsal scale reduction happens at the level of 22 (DSR 19 to 18), 25 (DSR 18 to 17), 34 (DSR 17 to 16) and 38 (DSR 16 to 15) ventral scales. On the underside of the body, two preventral scales, followed by 206 ventral scales. The anal plate divided, followed by 83 pairs of subcaudal scales, ending to a single terminal scale.

Coloration. The dorsal head ground color grayish-white, with a blackish blotch on the posterior prefrontals and anterior frontal, and a parenthesis-shaped blackish blotch on the parietals. Dorsal head scale sutures with irregular feebly blackish dots. On the sides of head, irregular blackish blotches scattered around eye, a blackish stripe running from posterior eye edge along the margin of the parietal on each side of the head. Snout and labial region whitish with irregular blackish dots adjacent to the eye. The underside of the head whitish. Dorsal body and tail ground color grayish white. Three blackish longitudinal stripes on the dorsal and lateral sides of the nape, changing to continuous black blotches on dorsal surfaces of body and tail. Dorsum with nearly parallel blackish dorsal bands, having irregular margins, the width of each band about one and a half of dorsal scale length, separated by a grayish-white interspace of about the length of one scale. Body sides with continuous blackish blotches alternating with dorsal bands. Dorsal blotches fade to scattered blackish spots posteriorly on dorsal surfaces of the tail. The ventral surface of the body whitish.

Cranial Osteology. The skull in Persiophis is long and elliptical and well ossified. At the tip of the snout, the single, pyramid-shaped premaxilla is deeply wedged in the space between the septomaxillae and the nasals. The nasals are directed downward. Left and right articulated nasals form a median septum between the nasal cavities and cover it dorsally. Ventrally the nasals form a process which lies in between the two frontals. Left and right septomaxillae are plate-like, bifurcate anteriorly and in contact medially. They form the floor of nasal cavity. Septomaxillae contact the nasal septum medially and posteriorly form a process that contact the frontals. The septomaxillae are partly fused with the vomers. The toothless vomers lie beneath the two septomaxillae and form a pair of spherical cavities in which lies the vomeronasal organs. The vomeronasal organs open by paired orifices into the buccal cavity. On each side of the head, a cone-shaped prefrontal borders the orbit anteriorly. Dorsally, the prefrontals have a tight articulation with the anterolateral surface of the frontals, and ventrally they bear a rather loose articulation with the maxillae.

The neurocranium is composed of compactly ossified bones, fully fused to each other to form a complete enclosure of the brain. Left and right frontals are well separated at the tip but joined together along the rest of their length. Parietals are ovally shaped, fused together to form a single bone (largest cranial element) that dorsally roofs the braincase, bearing no elaborated crests. Laterally it extends far down either side of the brain, reaching the basisphenoid and the prootics. Left and right postorbitals articulate with the anterolateral surfaces of the parietal and form the dorsoposterior boundary of each orbit.

Left and right prootics are quadrate shaped bones, partly fused with the parietal and forming the anterior walls of each internal otic capsule. They also constitute the anterior half of each fenestra ovalis and the posterolateral wall of the braincase. Left and right supraoccipitals are fused together to form a single bone. Externally it roofs the posterior brain cavity, internally it expands to form the posterior part of each otic capsule. A pair of diagonal crests extend transversally along the posterior part of the supraoccipitals. Left and right exoccipital bones form the posterolateral wall of the braincase, as well as part of its roof. They are fused with the opisthotics and together surround the jugular foramen and extend forward to form the posterior border of the fenestra ovalis. They form the entire oval foramen magnum, except for a small ventral portion of the occipital condyle. The basioccipital forms the floor of the posterior part of the brain cavity and the ventral portion of the occipital condyle. It completes the foramen magnum and creates a big, thick and raised occipital condyle. The basioccipital forms the posterior braincase floor. The basisphenoid and parasphenoid are fused to each other to form a single, long bone. It forms the posterior snout and anterior braincase floor.

In the palatomaxillary arches, the palatines are long and narrow, articulate with the prefrontal process of the maxilla laterally and with the pterygoid posteriorly. There are three small sized teeth at the mid-length of each palatine. The pterygoids are edentulous, long and bent bars that are narrow anteriorly, flattened posteriorly, and extend from the posterior palatines to the posterior mandibles. The ectopterygoids are flat, bifurcate anteriorly, notched posteriorly and connect the maxillae to the pterygoids. Left and right maxillae are curved, anteriorly thin, posteriorly somewhat broadened and connect to the flattened ventral surface of the ectopterygoid by a mesial process. The maxilla medially articulates with the ventral surface of the prefrontal. Each maxilla is edentulous anteriorly, bears three small teeth, and after another edentulous medial region, bears 6/5 small teeth. Finally, after a small space (equal to the length of one socket) two big, elongated, posteriorly curved teeth are present. In the medial, edentulous region of the left maxilla, a small socket is observed.

The mandibular units are composed of compactly ossified bone elements. Left and right supratemporals are narrow, flattened, dermal elements, connected to the proximal end of quadrates and the posterolateral part of braincase by fibrous connective tissue. Each supratemporal is long, slightly bent upward, and overlays the exoccipital, prootic and even reaches the edge of the parietal. Left and right quadrates are long, thick, rectangular shaped, having a flattened proximal end aligned along the posterolateral border of each supratemporal. The distal articulating surface of each quadrate is extended transversely and directed backward. Left and right mandibles are long, dorsally concave, connected to each other anteriorly by an elastic ligament. Each mandible unit is composed of two major bones, a compound bone and dentary. The dentary is somewhat dorsally curved and bears sockets for closely set 18/19 (L/R) small teeth that decrease in size posteriorly. Left and right stapes (columella) are slender, rod like bones, proximally enlarged and form a footplate that fit into the fenestra ovalis, distally connect to the inner surface of the quadrate at about mid length level.

Natural history. Our data on biology of Persiophis fahimii is based on the one specimen collected. The holotype was collected at elevation of 1,350 m ASL on a bare mountainside, while climbing on a vertical rocky wall, at late night (2.30 AM). The mountain is composed of Devonian limestone marbles, at the southeastern edge of the central mountains of Iran, ranging from 1,050 to 1,600 m ASL. Dominant vegetation on the plain in front of the mountain is Calligonum and annual forbs and grasses. At the base of the mountain, the vegetation changes to Calligonum and Ziziphus nummularia. At the type locality, the vegetation is dominated by sparse woody, thorny or aromatic shrubs, including Periploca sp. (Apocynaceae), Dichanthium sp. (Poaceae), Fagonia sp. (Zygophyllaceae), Ephedra foliata (Ephedraceae), Teucrium sp. (Lamiaceae), Lophochloa sp. (Poaceae), Lycium sp. (Solanaceae), Tribulus sp. (Zygophyllaceae), Pulicaria sp. (Asteraceae), Reseda sp. (Resedaceae), Heliotropium sp. (Boraginaceae), Gymnocarpos decander (Caryophyllaceae), Convolvulus sp. (Convolvulaceae), Heliantemum sp. (Cistaceae), and Diceratella persica (Apiaceae).

Conservation. Since the first field expedition in 2008 and two more field expeditions in spring 2017 and 2018 in the type locality of Persiophis fahimii failed to find any additional specimens of this snake, we assume that this snake is a very rare species with a limited local distribution. Currently, there is not enough data to evaluate the conservation status of Persiophis fahimii; hence, further expeditions are needed to shed light on the distribution and ecology of this snake. We suggest it be considered to have the IUCN Red List status ‘DD—Data Deficient.’ But researchers should take care in studying the species, avoiding over collecting or disturbing the habitat. We suggest that the local conservation management around the type locality of the species is urgently required.

Discussion

Additional information on Dolichophis andreanus (Werner, 1917) comb. nov.

Though Hierophis andreanus was originally described as Zamenis andreana by Werner (1917), it was an unknown and forgotten snake not listed in regional checklists (Latifi, 1991; Leviton et al., 1992) until researchers in the first decade of 21st century shed light on its distribution (Rajabizadeh & Rastegar-Pouyani, 2006; Rajabizadeh & Rastegar-Pouyani, 2009; Schätti, 2001). Since the genetic proximity of the species to dwarf snakes (genus Eirenis) and morphologic similarity to whip snakes (genera Hierophis and Dolichophis) were in contrast (Schätti & Monsch, 2004), its taxonomy was obscure and authors referred to it as Coluber (s. l.) andreanus (Rajabizadeh & Rastegar-Pouyani, 2006). Rastegar-Pouyani et al. (2008) erroneously listed this species in the genus Zamenis. Torki (2010) assigned the species to the genus Hierophis without any taxonomic justification. Surprisingly, other authors followed this classification without further questioning its taxonomic status (Chefaoui et al., 2018; Wallach, Williams & Boundy, 2014). Recent phylogenetic studies on snakes cast doubt on the taxonomic placement of Hierophis andreanus within the genus Hierophis (Figueroa et al., 2016). Our molecular phylogenetic results clearly indicate placement of Hierophis andreanus within the genus Dolichophis (Fig. 2), hence we suggest the new combination Dolichophis andreanus (Werner, 1917) comb. nov.

From a comparative point of view, the overall shape of skull and neurocranium in Dolichophis andreanus generally resembles that of Eirenis more than of Hierophis and Dolichophis (Fig. 6). In both D. andreanus and Eirenis, the neurocranium is wide, ovally shaped, bearing no elaborated V-shaped pair of crests on the parietal bones, and the braincase is large. The skull is long and elliptical, well ossified and composed of relatively thick bones. On the tip of the snout, there is a single, pyramid-shaped bone (premaxilla), that is dorsally wedged between the nasals, and like Eirenis, it is projected less anteriorly than in Hierophis and Dolichophis. Compared to whip snakes, the neurocranium in Dolichophis andreanus is wider, bearing a less-elaborated V-shaped pair of crests on the parietal bones, again resembling Eirenis. The CT-scanned Dolichophis andreanus specimen has 10/10, 9/9, 10/9, 15/13 curved teeth on maxilla, palatine, pterygoid and dentary bone.

Figure 6 Micro-CT reconstruction of the skull cranial osteology of Dolichophis andreanus comb. nov.

(A) Lateral and (B) dorsal views of the skull. Visualization by Mahdi Rajabizadeh.

Based on the head and body scalation data, Schätti & Monsch (2004) inferred a sister-group relationship between Dolichophis andreanus and dwarf snakes of the genus Eirenis, especially based on similar traits including the low number of supralabial, infralabial, anterior temporal and dorsal scale rows. Morphological similarity between Dolichophis andreanus and Eirenis is striking. The evolutionary history of head and dorsal body scales, as well as total size shows that the most parsimonious state for the common ancestor of Western Palearctic racers, whip snakes and dwarf snakes, is a large-size snake (total size more than one meter) having two anterior temporals, 8 supralabials, 9–10 infralabials and 19 dorsal scales. Total size of less than one meter, a single anterior temporal as well as 15, 17 and 18 dorsal body scales evolved independently in both Eirenis and D. andreanus. The number of supralabials and infralabials is not totally unique in dwarf snakes, hence 7–8 supralabials and 7–9 infralabials are present in the genus Eirenis and in D. andreanus as well.

Additional information on Muhtarophis barani (Olgun et al., 2007)

Although previous phylogenetic studies did not unambiguously resolve the phylogenetic position of Muhtarophis barani or Baran’s Black-headed Dwarf Snake (Avcı et al., 2015; Šmíd et al., 2015; Tamar et al., 2016), our tree surprisingly places it strongly as the sister group to the genus Scaphiophis. African Shovel-nosed Snakes (S. albopunctatus (Peters, 1870) and S. raffreyi (Bocourt, 1875) are large-sized snakes, maximum total length around 150 centimeters (Broadley, 1994), distributed around the periphery of the Central African rain forest from Ghana to western Ethiopia and adjacent Sudan (Largen & Rasmussen, 1993). In contrast, Muhtarophis is a dwarfed ground snake with maximum total length around 40 centimeters, reported from Hatay Province, Southern Turkey (Avcı et al., 2015).

The skull in Scaphiophis is robust, the premaxilla is large, beak shaped and divides the nasals, and each lateral projection of premaxilla is actually indeed divided into two lobes, the posterior nasals are articulated to the middle of the anterior frontal, the quadrate is not oblique nor slanting backward, dentition in a sample of S. albopunctatus is maxillary 15, palatine 9, pterygoid 8, dentary 18, and in a sample of S. raffreyi is 13, 7, 7, 16 respectively (Bourgeois, 1968; Broadley, 1994). In Muhtarophis, the skull is also robust, having a large pyramid shaped premaxilla that is wedged between the anterior nasals. The posterior nasal is broadly articulated to the anterior frontal, the quadrate is more or less vertical, and the dentition in two examined samples consists of six maxillary 6 heterogeneous teeth (5 same size anterior teeth and one about two times larger rear tooth), palatine 4, pterygoid 8, dentary 9 (Avcı et al., 2015). Though there are some shared osteological traits between Scaphiophis and Muhtarophis, the obvious differences in osteology of these genera makes a sister group relationship doubtful, despite the strong support estimated here. The variable placement among the phylogenetic analyses may result from a lack of taxon sampling, or more likely, inadequate sampling of independent loci and phylogenetically informative molecular characters. Hence, further research is needed to identify the phylogenetic position of Muhtarophis.

The challenge of monotypy

Since currently only one species is known in Persiophis, the genus is monotypic. Moreover, the sole species is known only from a single specimen, which is a common problem in squamate taxonomy (Meiri et al., 2018). Two scenarios exist which may, in the future, avoid the challenge of monotypy and demonstrate monophyly of the genus based on cladistic theory, which generally demands that a genus is a monophyletic group of species; thus, monotypic genera are not phylogenetically informative (Platnick, 1976). First, since the reptile fauna (especially snakes) of southwestern Asia is not sufficiently studied and many undescribed taxa still likely remain (Rajabizadeh, 2018), it is possible that other species within the genus Persiophis exist that have not been discovered to date, either extant or in the fossil record. Regardless, since the species Persiophis fahimii is strongly supported as a lineage distinct from any existing snake genera, based on the molecular phylogeny and osteological analyses, we here accept it as a representative of a distinct genus Persiophis that is currently monotypic.

Conclusions

Here, we present new molecular sequence data and a new phylogenetic analysis of snakes, focusing primarily on Colubrinae from southwestern Asia. We find continued uncertainty in the placement of the enigmatic Turkish genus Muhtarophis based on osteological comparisons, despite strong support in the phylogenetic analysis. On the basis of the tree and morphology, we confidently reassign Hierophis andreanus from Hierophis to Dolichophis, hereafter referred to Dolichophis andreanus. Our morphological and molecular data also suggest a potential instance of convergent miniaturization in these Old-World racers. Finally, we report on the discovery of a new genus and species of ground snake, Persiophis fahimii, from southeastern Iran. Our data highlight the importance of broad phylogenetic sampling and ground-level field research to gather an accurate picture of global biodiversity, phylogenetic relationships, and evolutionary patterns in groups such as snakes.

Supplemental Information

Table S1 Details of the new sequences of the mitochondrial DNA (12S rRNA, ND4, and CYTB) and nuclear DNA (RAG2, NT3, CMOS, and BDNF) genes and accessioned GenBank numbers

Click here for additional data file.

Table S2 Number of projections and voxel size of the scanned specimens

Click here for additional data file.

Table S3 Selected morphological characters of Colubrine snake genera of India, Central and Western Asia and Northern Africa

Abbreviations: Pt, Pterygoid teeth (+ = present, 0 = absent); Pr, Premaxilla (N = Normal, P = projected); Mx, Large diastema on the middle maxilla (+ = present, 0 = absent) [ops = opisthoglyphous]; T, Tooth shape (N = Normal, V = small and vestigial); 1L, First supra labial scale (≤ = smaller or eqqual to the second supra labial, > bigger than the second supra labial); R, Rostral scale (N = Normal, P = Projected); Dor; Number of dorsal scales at midbody; Pupil (R = round, V = vertical, H = horizontal). Data are based on (Boulenger, 1898); (Bourgeois, 1968); (Broadley, 1994); Broeckhoven & du Plessis, 2017; (Chan-Ard, Nabhitabhata & Parr, 2015); (Chippaux & Jackson, 2019); (Das et al., 2019); (Gans, 1954); (Kardong, 1979); (Kharin & Akulenko, 2008); (Mahlow et al., 2013); (Marx, 1959); (Mirza & Patel, 2018); (Mirza et al., 2016); (Nguyen et al., 2020); (Poyarkov, Nguyen & Vogel, 2019); (Rajabizadeh, 2018); (Saleh & Sarhan, 2016); Schätti, 1985, 1987; (Tsai & Mao, 2017); (Utiger, Schätti & Helfenberger, 2005); (Wade, 2008); Wagner & Böhme, 2007; (Wallach, Lanza & Nistri, 2010); (Wang, Shi & Guo, 2019).

Click here for additional data file.

Figure S1 The holotype of Persiophis fahimii Gen. et sp. nov., preserved specimen

(a) Dorsal and (b) ventral views of the whole body; details of head scalation in close-up (c) dorsal, (d) lateral and (e) ventral views. Photos by Dominique Adriaens.

Click here for additional data file.

Data S1 Newly obtained mitochondrial DNA and nuclear DNA sequence data of Colubridae snakes

Click here for additional data file.

We are grateful to Eskandar Rastegar-Pouyani for his kind assistance in the field and reviewing the manuscript. Thanks to Firouzeh Bordbar (Bahonar University, Kerman) for identification of plant species. We are also grateful to Barbara De Kegel (UGent), Jens Vindum and Lauren Scheinberg (CAS) and Ted Papenfuss (MVZ) for their assistance. Also we thank Daniel Melnikov, Hossein Nabizadeh and Morteza Moaddab for their help in the field. We thank Justin L. Lee (NMNH) and an anonymous reviewer for their useful suggestions which helped us to improve the previous version of our manuscript.

Additional Information and Declarations

Competing Interests

Author Contributions

Animal Ethics

Field Study Permissions

Data Availability

New Species Registration

Nikolay A. Poyarkov serves is an Academic Editor for PeerJ. The other authors declare that no conflicts of interest exist.

Mahdi Rajabizadeh and R. Alexander Pyron conceived and designed the experiments, performed the experiments, analyzed the data, prepared figures and/or tables, authored or reviewed drafts of the paper, and approved the final draft.

Roman Nazarov conceived and designed the experiments, analyzed the data, prepared figures and/or tables, authored or reviewed drafts of the paper, and approved the final draft.

Nikolay A. Poyarkov, Dominique Adriaens and Anthony Herrel performed the experiments, analyzed the data, prepared figures and/or tables, authored or reviewed drafts of the paper, and approved the final draft.

The following information was supplied relating to ethical approvals (i.e., approving body and any reference numbers):

Specimen collection protocols and animal operations followed the Institutional Ethical Committee of International Center for Science, High Technology and Environmental Sciences, Kerman, Iran (Certificate #1.87-1).

The following information was supplied relating to field study approvals (i.e., approving body and any reference numbers):

Fieldworks, including the collection of the samples and animals in the field, was performed outside of any protected area, in the framework of a project contract signed by the International Center for Science, High Technology and Environmental Sciences, Kerman, Iran (contract number 1.87, issued at 11.04.2008). The contract bears permission to collect the reptile samples outside of any protected area of the Department of the Environment that needed extra permissions.

The following information was supplied regarding data availability:

Voucher specimens along with tissue samples examined in this study were preserved and deposited in the International Center for Science, High Technology and Environmental Sciences Herpetological Collection (ICSTZM), Kerman, Iran:

- ICSTZM.7H.1154 Hierophis andreanus (Darreh Shahr City, Ilam Province, Iran)

- ICSTZM.7H.1151 Persiophis fahimii Gen. et sp. nov. (Orzueeyeh City, Kerman Province, Iran). 

The exact geographic localities of the type locality of the new species are not given herein due to concerns regarding the conservation of the species. However, this information can be obtained from authors upon request.

The sequences are available in the Supplemental Files and the sequences of ND4, CYTB, 12S, RAG2, NT3, CMOS and BDNF genes are available at GenBank: MN531564, MN531565, MN536808, MN531566, MN531567, MN536809, MT163746, MT163747, MT163748, MT163749 (Table S1).

The following information was supplied regarding the registration of a newly described species:

Publication LSID:

urn:lsid:zoobank.org:pub:4EAACC14-8FC8-46B9-830C-4AEC8A42A562

Persiophis

LSID: urn:lsid:zoobank.org:act:31BFF5E6-D2F9-4BAA-816A-99DEB8E45F65

Persiophis fahimii

LSID: urn:lsid:zoobank.org:act:DDF27FFC-6C55-4767-9627-0FFB2F16155E

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
