# Peer review of "Additions to the phylogeny of colubrine snakes in Southwestern Asia, with description of a new genus and species (Serpentes: Colubridae: Colubrinae)"

_PeerJ, doi:10.7717/peerj.9016_

## Round 0.1 · original submission · Major Revisions

Although both reviewers found merit in your study, they found a number of issues that need to be addressed. I'd ask you to pay particular attention to the comments by Reviewer 2.

·

Basic reporting

The paper’s English is well-written and professional. The authors show subject-level expertise in Iranian Herpetology and provide sufficient literature references. All contents of the paper are relevant to their original research questions. However, some of the figures and references section need re-formatting.

Experimental design

The paper fits the aim and scope of the journal well, given PeerJ’s history of herpetological submissions dealing with systematics. I have no objections to their research questions and analyses used. Their methods all seem repeatable. Some of the descriptive information is placed in the results portion of the paper (i.e. how ventral scales were counted) and should be moved to the methods section.

Validity of the findings

It is clear that the authors have a new taxon. The materials and paperwork they provided appear authentic.

Additional comments

The article "Additions to the phylogeny of colubrine snakes in Southwestern Asia, with description of a new genus and species (Serpentes: Colubridae: Colubrinae)" provides an important contribution to the systematics of Western Palearctic Colubrid Snakes and should be accepted after the authors complete minor revisions. The authors of this paper have clear evidence for the description of a new genus/species of snake and for the transfer of Hierophis andreanus to the genus Dolichophis. However, there are a few key issues that the authors should address in the manuscript before it is slated for publication.

I offer the following broad suggestions:

1. The generic diagnosis and comparison of Persiophis fahimii Gen. et sp. nov. is completely based on osteological data. The authors should also include external characters in these sections as well to aid in field identification (i.e. pholidosis + color pattern). This would significantly improve the generic diagnosis.

2. Methods on how morphological characters were counted for the description of Persiophis fahimii Gen. et sp. nov. should be moved to the Materials and Methods section of the paper. Some important information is also missing in the materials and methods (i.e. How was the tissue collected from these specimens? How are the tissues preserved?).

3. The English name Andreas’ Racer is used throughout the paper. This should be changed in most instances to its scientific name so it reads as Coluber (s. l.) andreanus and then later Dolichophis andreanus comb. nov. The current generic position of the species is in Hierophis so the authors should also justify in the introduction why they use Coluber instead.

4. Some of the figures for ancestral state reconstructions (Figs. 7–9) should be edited and redone in higher resolution so they look more professional.

There are other minor issues/suggestions that should be addressed by the authors as well, which I have noted in my comments on the attached PDF. I also cleaned up some of the references in their bibliography so they are consistent in formatting.

– Justin L. Lee

Reviewer 2 ·

Basic reporting

1. Missing literature pointed out in my comments

2. Additional tables required, some raw data (measurements & counts) not provided (Figures 7-9 tabulate raw data in supplementary material), photographs of the preserved specimen and its condition. Photographs, measurement and counts for Andreas’ Racer (add in supplementary material). Provide a GenBank table for the sequences used in this study at least for the "Old world racers"


Check general comments section and suggestions in my review for more details.

Experimental design

Comparative morphology sections (external morphology and osteology) needs improvement. It is incomplete now. Therefore not a rigorous investigation.

Check general comments section and suggestions in my review for more details.

Validity of the findings

The new genus is described based on one female specimen because it is rare which is understood. Therefore, it is the responsibility of the authors to do a thorough comparison with all the genus in the lineage and all the closely related sister species.

Some claims ("smaller head elements may be accompanied by a lower number of covering scales including labial and temporal scales") in the discussion section are not statistically sound. Therefore, I suggest these to be excluded or do a thorough comparisons to back these claims.

Check general comments section and suggestions in my review for more details.

Additional comments

This manuscript require a lot of improvements. I have made my comments and suggestion in the manuscript. These are some of the most important components to be dealt with in this manuscript.

Molecular: 1. Mitochondrial genes tend to have faster rate of molecular evolution and therefore higher genetic difference. I think it is important to add at least one nuclear gene in this study, 2. I see you have done only a ML analysis and the bootstrap support is "80" for the new genus and its closest sister, while this support is "89" for Scaphiophis and Muhtarophis but you have less confidence on the phylogenetic placement of the latter pair than the former. Therefore, it is better you run a bayesian analysis for the “old world racer” dataset to have more than one line of evidence to support this relationship. 3. Recent studies shows that Argyrogena is also within the old world racer group add this taxa to your phylogeny

Morphology: 1) Provide detailed morphological data table for the new taxa and figures of the preserved holotype 2) Provide a comparative osteology table for all old world racer genera, apart from this compare osteology of the new genus with all the sister species in the genus (Rhynchocalamus, Wallophis, Wallaceophis and Lytorhynchus). 3) Provide a comparative external morphology table for all old world racer genera.

Ancestral state estimation is incomplete, needs further research: 1) Add information for all the old world racers and then do a ancestral state estimation if not tabulate the scalation data. 2) If your idea is to show that dwarfism has evolved multiple times in this group do so by showing them in the phylogeny of all the old world racers.

Annotated reviews are not available for download in order to protect the identity of reviewers who chose to remain anonymous.

---

## Round 0.2 · Minor Revisions

I recognize that you made and effort to accommodate the comments by the reviewers, but there are two issues raised by Reviewer 2 that I'd like you to address further.

·

Basic reporting

Writing and grammar is improved from previous draft, references section has been cleaned up and is more consistent.

Experimental design

See first draft for comments, experimental design is sufficient for publication

Validity of the findings

Discovery of new genus/species is certainly valid, other taxonomic changes seem supported by phylogenetic data.

Additional comments

The authors have made improvements on this new manuscript and have addressed the suggestions of my first review adequately. There are some small changes I wish they fix, but I would not go so far to suggest Minor Revisions. My comments are attached as a PDF file.

Reviewer 2 ·

Basic reporting

no comment

Experimental design

Some of the analysis I suggested to exclude in the previous revisions are retained although the data is incomplete. I suggested to add 'one' nuclear gene to their dataset, which is not included.

Validity of the findings

The new genus and species as such is valid and the authors have added additional comparative morphological data which I had requested in the previous review.

Additional comments

I appreciate the authors for addressing most of my minor comments and some of my major comments. However, I still have the two major issues in this MS which is not addressed.

1. Additional genetic data was not added. At this day in age when the costs of DNA sequencing are cheap and the authors have cited lack of funds to add “one” additional nuclear gene sequence is unacceptable. I would not be this particular if it was only a species description. The authors are also trying to describe a whole new genera here hence this data is a must

2. I asked the authors to remove sections on the examples of parallel evolution because this is carried out based on an incomplete dataset therefore misleading. I still think this section will be misleading without a complete dataset. I suggest them to remove this.

---

## Round 0.3 · accepted · Accept

I believe that you properly addressed all of the remaining issues raised by the reviewers.